# Study on the Autonomous Walking of an Underground Definite Route LHD Machine Based on Reinforcement Learning

**Shuo Zhao** [1], **Liguan Wang** [1,2], **Ziyu Zhao** [1] **and Lin Bi** [1,2,*]

1 School of Resources and Safety Engineering, Central South University, Changsha 410083, China; 205512125@csu.edu.cn (S.Z.); liguan_wang@163.com (L.W.); csu_zzy_93@163.com (Z.Z.)
2 Changsha Digital Mine Co., Ltd., Changsha 410221, China
* Correspondence: mr.bilin@csu.edu.cn

**Abstract:** The autonomous walking of an underground load-haul-dump (LHD) machine is a current research hotspot. The route of an underground LHD machine is generally definite, and most research is based on the logic of positioning-decision control. Based on a reinforcement learning algorithm, a new autonomous walking training algorithm, Traditional Control Based DQN (TCB-DQN), combining the methods of traditional reflective navigation and reinforcement learning deep q-networks (DQN), is proposed. Compared with the logic of location-decision control, TCB-DQN does not require accurate positioning, but only determines how to reach the endpoint by sensing the distance from the endpoint. Through experimental verification, after using the TCB-DQN algorithm for training in a simple tunnel, the LHD machine could achieve a walking effect similar to that of a human driver's manual operation, while after training in a more complex tunnel, the TCB-DQN algorithm could reach the endpoint smoothly.

**Keywords:** virtual simulation; reinforcement learning; LHD machine control





## 1. Introduction

As the main equipment for underground transportation and loading in non-coal mines, the efficiency of LHD machines directly affects the productivity of mines. Traditionally, a LHD machine needs to be driven and controlled by professionals. Although this can achieve a good result, the working environment of an underground LHD machine is poor, and problems such as dust, noise, and vibration are common, which have a serious impact on the driver's health. Therefore, the main development trend of underground unmanned technology has become making an LHD machine achieve autonomous walking and shovel loading [1–3].

The maintenance and use cost of an actual LHD machine is high. Frequent field experiments consume a lot of human and material resources and affect the normal production process. With the development of simulation software, such as Bullet, Mujoco, Carla, V-rep, and the Unreal Engine, it is possible to more accurately simulate various situations from reality on a computer. The Unreal Engine is a game engine developed by Epic Games in 1998. Unreal Engine 4 was launched in 2014, and the source code is open source. UE4 has a complete set of development tools. It is widely used in automatic driving, military training, safety training, etc. Michalík [4] developed a set of driving simulation platforms based on UE4, to analyze the driving safety of drivers, while Sim Centric and Torch Technologies have developed a VR system for tactical training in cooperation with military institutions. Carla, an automatic driving simulation platform, is another a secondary development based on UE4. Therefore, UE4 is very reliable and effective as a platform to build an autonomous walking simulation environment for an LHD machine.

The underground LHD machine is a special vehicle, which is essentially an articulated vehicle, that is, a vehicle composed of two or more car bodies connected by a hinge device. Its special structure creates a unique steering mode [5]. The dynamic model of an articulated

vehicle can be divided into two degrees of freedom [6], three degrees of freedom [7], four degrees of freedom [8], and multiple degrees of freedom [9,10]. Theoretically, four degrees of freedom can meet the needs of driving force distribution control and autonomous driving control. Bai [11], of Beijing University of science and technology, optimized a model based on the classical articulated vehicle kinematics model [12] proposed by Corke et al., and regarded the articulated vehicle as a rigid body, ignoring the tire slip, and established a set of dynamic models suitable for a four-wheel independent drive articulated vehicle.

The research on LHD machine autonomous walking mainly focuses on the research of control algorithms, such as PID (proportion integral differential), LQR (linear quadratic regulator), SMC (sliding mode control), MPC (model predictive control), etc. Aslam [13] et al., based on the SMC [14] algorithm and inspired by the research of SSV (skid-steer vehicles) [15–18] based on FLC (fuzzy logic control), integrated FLC [19,20] with SMC to solve the jitter problem often encountered in traditional sliding module control. Jiang [21], based on the PID algorithm and the reactive navigation technology of "along the wall", designed an autonomous navigation bivariate PID controller to realize the autonomous navigation and driving of a LHD machine. Nayl [22,23] and others established a lateral control algorithm based on MPC control for a complete kinematic model of an articulated vehicle. Although the vehicle cannot reach the speed of manual operation, it can stably realize the autonomous walking of the LHD machine. Wu [24] applied the QPSO (quantum-behaved particle swarm optimization) algorithm to the selection of LQR control weighting matrix parameters and proposed the LQR-QPSO algorithm, which greatly shortens the time for correcting the body attitude of an underground LHD machine in case of abnormal conditions. Benefiting from the rapid development of computer performance in recent years, machine learning has been widely used in the field of vehicle control [25–27]. Kanarachos Stratis [28,29] used a feedforward neural network to track the trajectory of intelligent vehicles and achieved good experimental results. Shao et al. [30–33] proposed to use of the reinforcement learning method to adjust PID control parameters in real-time, and applied this to the autonomous walking of an articulated vehicle. Experiments showed that, under the support of reinforcement learning, a PID controller can effectively reduce vibration. Although these control algorithms can achieve good results, this depends on the artificially given route, and the use of a reinforcement learning algorithm will enable the LHD to explore a route by itself; so as to realize the real sense of autonomous walking.

In the working scene of an LHD machine, the state space is mainly the vehicle speed, articulation angle, body surrounding environment information, etc. Through reinforcement learning, an LHD machine can independently learn walking control in a complex environment. However, relying solely on reinforcement learning will make the LHD machine explore many unnecessary situations and reduce the training efficiency. Therefore, this paper proposes a training network, Traditional Control Based DQN (TCB-DQN), which combines a traditional control algorithm and DQN reinforcement learning method to train an LHD machine in autonomous walking. Its main contributions are as follows:

(1) An underground simulation LHD machine operation environment was built in UE4, based on realistic standards. The simulation environment can output a variety of information about the LHD machine itself, such as speed, steering angle, etc., as well as interactive information regarding the LHD machine and the surrounding environment, such as the distance between the vehicle body and the edge of the tunnel, the distance between the vehicle and the starting and ending points, etc. In this environment, the autonomous walking of LHD can be studied efficiently.

(2) A new training framework TCB-DQN is proposed, which combines traditional relative navigation and a DQN neural network. The framework adopts a weak constraint strategy, to avoid frequent route correction of the LHD machine. Compared with the traditional control algorithm, TCB-DQN changes the LHD autonomous walking mode from passive acceptance to autonomous exploration. In addition, compared with the DQN algorithm training, TCB-DQN gives the LHD machine the ability to judge when to start exploration, avoiding the LHD machine exploring in unnecessary situations,

greatly improving the training efficiency of the LHD machine, and enabling the LHD machine to still realize autonomous walking in a more complex environment.

The structure of this paper is as follows: Section 2 introduces the construction process of the simulation environment; in Section 3, the experimental design of different algorithms is carried out. In Sections 4 and 5, the algorithm is tested and analyzed, and the corresponding summary is made.

## 2. Tunnel Model Construction

The construction of the underground simulation platform was mainly divided into tunnel modeling and LHD machine model construction. Due to the high-level physical simulation effect and realistic rendering effect of UE4, UE4 was used to build the simulation platform. Tunnel and LHD machine models preprocessed from other modeling software were imported into UE4 and given corresponding material and physical properties, so that the interaction between models can be realized and the collision effect between models can be truly simulated.

### 2.1. Tunnel Simulation Modeling

The original 3D tunnel model was constructed using a ZEB-Horizon 3D laser scanner developed by GeoSLAM, which scanned and reconstructed the data in a real tunnel. The reconstructed point cloud data after scanning is shown in Figure 1a; after it was imported into UE4, the tunnel model was saved in the assets in the format of a static mesh, as shown in Figure 1b. The internal state of the initial tunnel is shown in Figure 1c; it can be seen that the ground of the tunnel model is extremely rough, which is too different from the flat ground in an actual transportation tunnel, which is related to factors such as ponding in the tunnel and stacking of obstacles during data collection. Therefore, the floor of the tunnel ground was flattened in UE4, and the final tunnel is shown in Figure 1d.

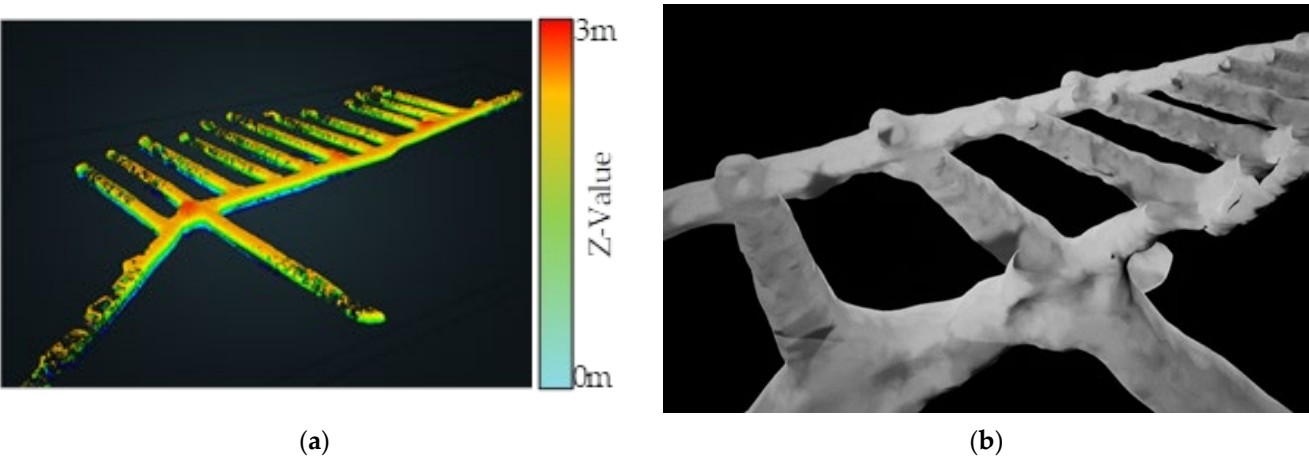

(**a**)                                   (**b**)

**Figure 1.** *Cont.*

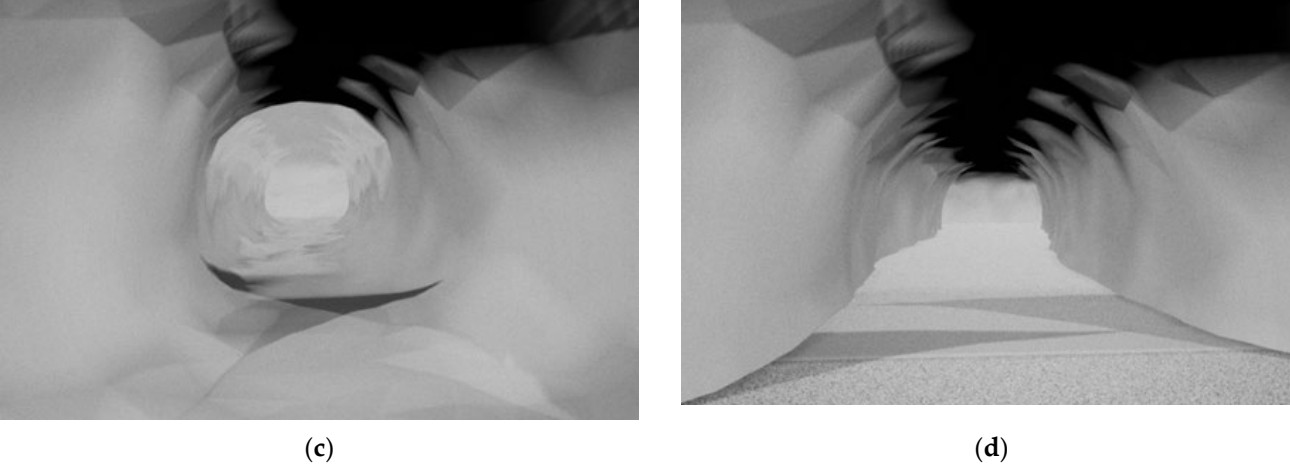

**Figure 1.** Tunnel point cloud data acquisition and model optimization, color represents the Z value. (**a**) Original point cloud data of real tunnel; (**b**) tunnel model after import UE4; (**c**) original state inside of the tunnel; (**d**) final state inside of the tunnel.

*2.2. LHD Machine Simulation Modeling*

The body of the LHD machine model is composed of front and rear parts, and the middle is connected by a hinge device. The bucket is divided into six Z-type reverse bars and eight forward, bars according to the structural characteristics. Based on the model of the LHD machine made in Qingdao Zhonghong, this paper selected six Z-type reverse bars as the structural characteristics to build the model. The main components of the LHD machine are the front body, the rear body, and the bucket. The bucket is connected to the front body by the tilting oil cylinder and piston, the lifting oil cylinder, piston, the swing arm, rocker arm, and connecting rod. To achieve steering of the LHD machine, two symmetrical steering pistons need to be set at the connection between the front body and the rear body.

UE4 has two methods to realize object displacement and rotation: physical simulation, and coordinate transformation. Using the physical simulation method requires turning on the simulated physics of the static mesh in advance and add constraints in different directions, it requires a high accuracy and no overlap between the various components, otherwise unpredictable errors will occur. Pure physical simulation is usually used when the model structure is not complex or there are few connections between models. For a complex body model, such as the LHD machine model, the use of pure physical simulation will not only increase the consumption of computer resources but also lead to various abnormalities.

For the movement simulation of the LHD machine, only the acceleration and deceleration control needs to be realized, and the overall coordinate transformation of the LHD machine can be realized through a physical formula, which can accurately simulate the walking of the LHD machine. Note that UE4 adopts a left-hand coordinate system instead of a right-hand coordinate system, as shown in Figure 2.

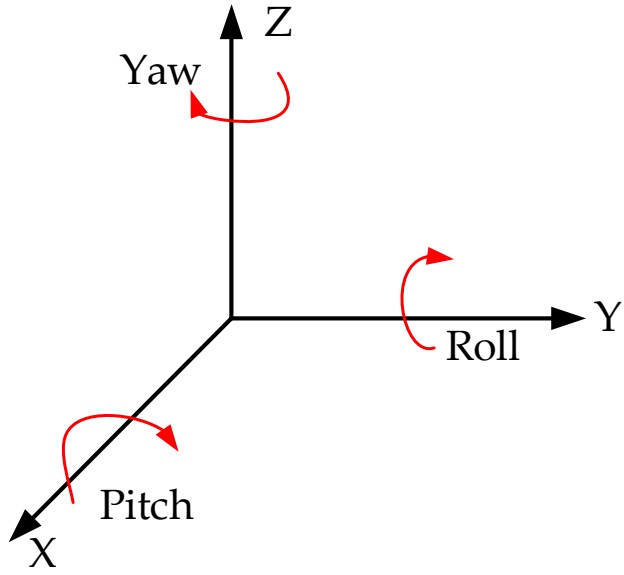

**Figure 2.** The coordinate system in UE4.

The movement of the LHD machine can be regarded as the movement of object coordinates in three-dimensional space, and the motion simulation of object coordinates is based on the translation and rotation transformation of the coordinate system. The translation transformation process can be simply represented by Equation (1):

$$\begin{cases} x_1 = x + t_x \\ y_2 = y + t_y \\ z_3 = z + t_z \end{cases} \tag{1}$$

The rotation matrix can be derived from the following matrix, which rotates $\alpha$ degrees around the Z-axis $R_Z$ is

$$R_Z = \begin{bmatrix} \cos\alpha & -\sin\alpha & 0 \\ \sin\alpha & \cos\alpha & 0 \\ 0 & 0 & 1 \end{bmatrix} \tag{2}$$

Similarly, rotates $\gamma$ degrees around the X-axis $R_X$ and rotates $\beta$ degrees around the Y-axis $R_Y$ are

$$R_X = \begin{bmatrix} 1 & 0 & 0 \\ 0 & \cos\gamma & -\sin\gamma \\ 0 & \sin\gamma & \cos\gamma \end{bmatrix} \tag{3}$$

$$R_Y = \begin{bmatrix} \cos\beta & 0 & \sin\beta \\ 0 & 1 & 0 \\ -\sin\beta & 0 & \cos\beta \end{bmatrix} \tag{4}$$

Therefore, the final rotation matrix of the coordinate system after three rotations of the object is:

$$\begin{aligned} R_{ZYX} &= R_Z R_Y R_X \\ &= \begin{bmatrix} \cos\alpha\cos\beta & \cos\alpha\sin\beta\sin\gamma - \sin\alpha\cos\gamma & \cos\alpha\sin\beta\cos\gamma + \sin\alpha\sin\gamma \\ \sin\alpha\cos\beta & \sin\alpha\sin\beta\sin\gamma + \cos\alpha\cos\gamma & \sin\alpha\sin\beta\cos\gamma - \cos\alpha\sin\gamma \\ -\sin\beta & \cos\beta\sin\gamma & \cos\beta\cos\gamma \end{bmatrix} \end{aligned} \tag{5}$$

Matrix $R_{ZYX}$ is the rotation matrix after the object rotates around the Z, Y, and X axes, in turn. According to the above method, the rotation matrix of the object according to other rotation orders can be solved. To avoid the problem of Gimbal lock, the quaternion method is introduced to correct it.

Referring to the dynamic model of a four-wheel independent drive articulated vehicle by Bai [11], as shown in the Equation (6), a set of dynamic models based on the articulated angle was established in the UE4 blueprint, and the turning radius and steering center of the front frame and rear frame were calculated in real-time using the articulated angle. When the simulation LHD machine has power input (i.e., the user operates the LHD machine through the key), the LHD machine will move along the track circle. The dynamic parameters of the simulated LHD machine are shown in Table 1.

$$
\begin{cases}
\dot{x}_f = v_f \cos \theta_f \\
\dot{y}_f = v_f \sin \theta_f \\
\dot{\theta}_f = \frac{v_f \sin \gamma + \dot{\gamma} L_r}{L_f \cos \gamma + L_r} \\
\dot{\gamma} = \dot{\gamma}
\end{cases}
\tag{6}
$$

**Table 1.** Dynamic parameters of the simulation LHD machine.

| Acceleration | Deceleration | Steering Angular Speed | Max Steering Angle | Max Velocity |
|:---:|:---:|:---:|:---:|:---:|
| $1 \, \text{m/s}^2$ | $1.5 \, \text{m/s}^2$ | $20°/\text{s}$ | $\pm 38°$ | $\pm 6 \, \text{km/h}$ |

Through the above description of the motion state of the space object and the mechanical model of the LHD machine in UE4, a blueprint was used to establish the function for calculating the object rotation matrix, and the simulation control of the LHD machine was realized through a matrix operation. The movement of the simulation LHD machine was realized by calling the set scene position and set scene rotation in the UE4 blue diagram every frame. For the construction of radar, the built-in function module BoxTraceByChannel can monitor whether there are objects within the specified range and feed the monitoring information, including the distance between the emission point and the object, to complete the simulation of the radar function.

### 2.3. Interaction Design

The underground simulation platform developed based on UE4 is an independent process and cannot directly interact with Python files using a third-party library. The reinforcement learning algorithms are based on Python's Pytorch library. To realize the interaction between our algorithm and UE4 simulation components, UE4 is regarded as a black box, and Pyautoigui transmits operation instructions to the simulation platform, to realize the operation of the LHD machine, the current state information of the LHD machine is obtained, and saving through the SaveStringToFile function in UE4 C++ library. Other operations such as calculating reward functions and selecting action instructions are implemented in Python. The main flow of interactions is shown in Figure 3.

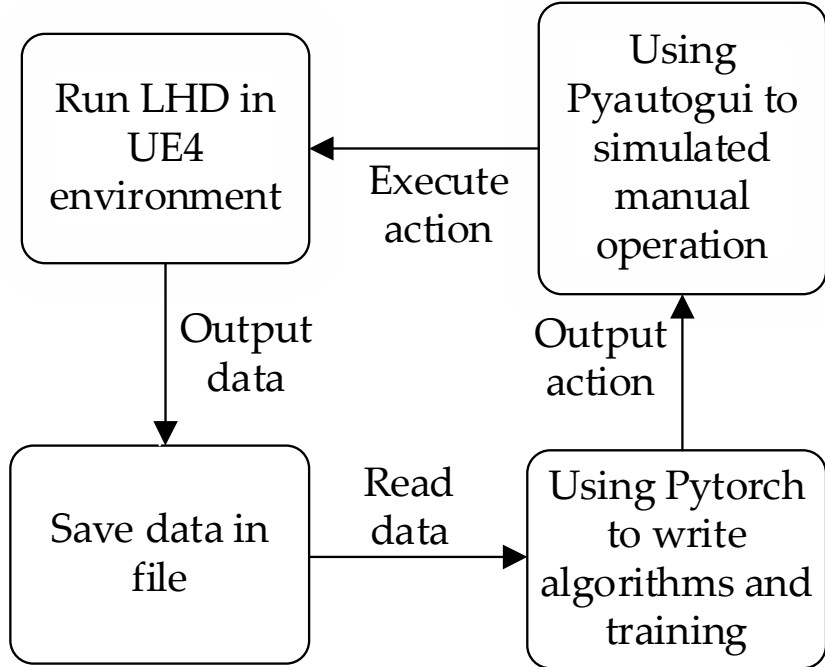

**Figure 3.** UE4 and Python interaction process.

### 3. Experimental Design of the LHD Machine Autonomous Walking

In general, the driving route of the underground LHD machine is fixed. Through the training of the LHD machine through reinforcement learning, the autonomous driving of the LHD machine on a fixed route can be realized. Reinforcement learning belongs to a branch of machine learning. When an agent interacts with the environment, the environment feeds back a reward to the agent, which can be positive or negative, large or small. This is determined according to the state change caused by the behavior of the agent in the environment. Reinforcement learning is based on the Markov decision process, combined with neural networks and other algorithms. Compared with supervised learning and unsupervised learning, it has a stronger universality. As shown in Figure 4, the agent takes action and executes it according to the reward $r$ in the environment. The environment returns the reward $r$ to the agent, according to the reward function. After the agent executes the action $a$, the state changes to $s$, and the cycle iterates. Each time, the action is selected according to the maximum reward. The configuration of the computer used in the experiment is shown in Table 2.

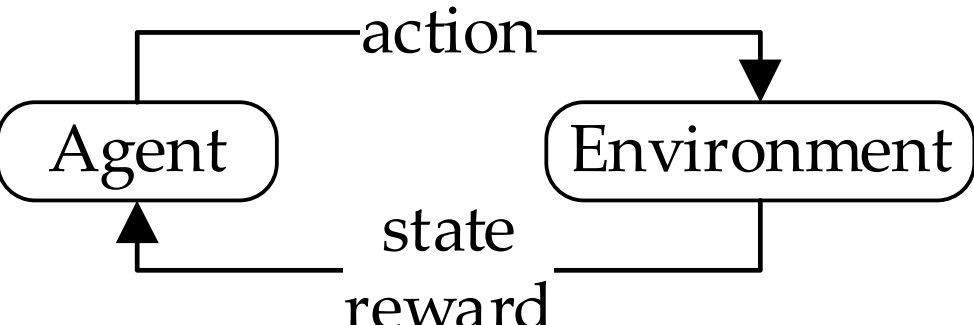

**Figure 4.** Reinforcement learning diagram.

**Table 2.** The hardware parameters.

| Hardware Name | Type |
|---|---|
| CPU | Intel(R) Core(TM) i7-10700 K @ 3.80 GHz |
| Graphics card | NVIDIA GeForce RTX 2060 SUPER |
| Memory | SAMSUNG 16 Gb@2666 MHz*2 |
| Hard disk | SAMSUNG 980 EVO 512 G |

*3.1. Using DQN to Realize the Autonomous Walking of an LHD Machine*

DQN plays a very important role in the value-based algorithm of reinforcement learning. It skillfully combines reinforcement learning with a neural network, to make up for the problem that the Q-table of Q-learning cannot be dynamically expanded. In addition, the concepts of experience reply and fixed target are introduced. One is used to store the experience of the agent. The other is used to improve the stability of training. It has been proven that DQN can achieve better performance than human beings in some fields [34]. Upadhay [35] and others achieved good results in using computer vision to realize the autonomous walking of vehicles; however, in the underground environment, it would be very difficult to use computer vision, due to the influence of dust, fog, and light. Oscar [36] used the DQN algorithm to realize the autonomous walking of vehicles in Carla, but added continuous points on the route to restrain the vehicles, which is tantamount to drawing up a route in advance. LHD machines need to be able to explore a path independently, so only the starting point and end point were given in the experiment. Whether using a Q-table or neural network, the purpose is to obtain the reward value of different actions in different states. First, take a specific state as the input layer parameter, and the output is the reward value of different actions in this state, after sorting the reward value, select the action in the first place. Figure 5 is the neural network model established in the DQN algorithm. That is, the state parameters of the LHD machine were input to the neural network, which can feed the possible reward for taking different actions in the current state, so that no error is generated due to discretization. The execution time of each action was set to 1 s, while the angular speed of the scraper was 20° per second, so a turning action can change the articulation angle by 20°.

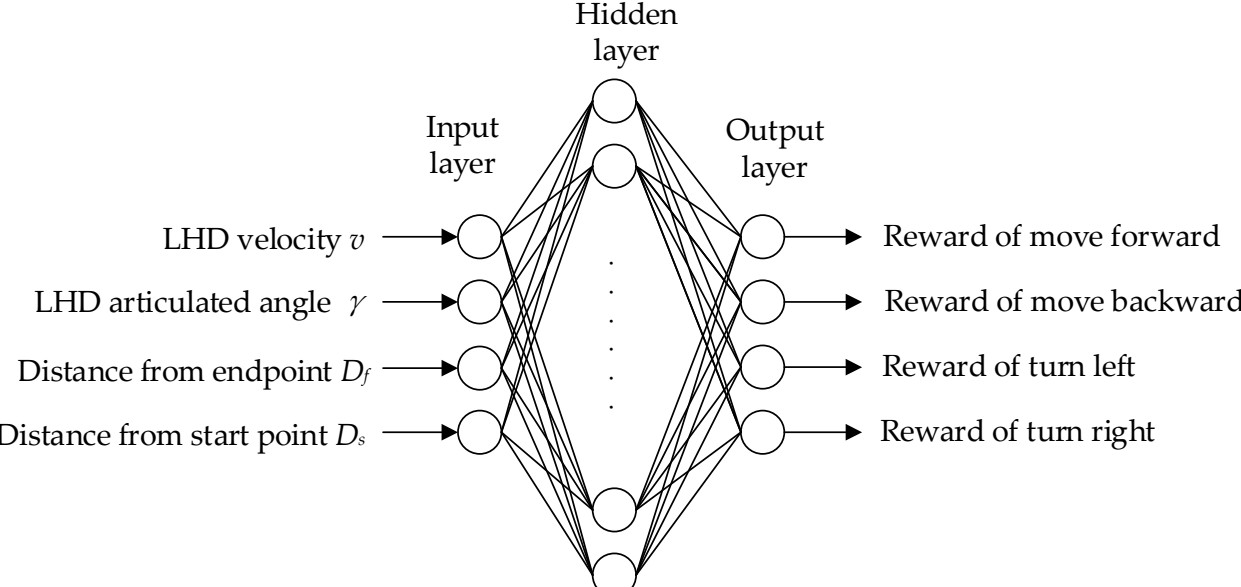

**Figure 5.** The structure of the neural network used in the DQN; input states of LHD machine, it can output the reward of different actions.

### 3.1.1. Hidden Layer Number Determination Experiment

One of the more important parameters in a DQN is the number of neurons in the hidden layer of the neural network. To achieve the best effect of the algorithm, experiments are carried out on the number of neurons at the hidden layer of different neural networks. If the number of neurons is too large, the over-fitting phenomenon is likely to occur after training, and the universality decreases. However, if the number is too small, the phenomenon of under-fitting can appear, which is of no practical significance for guiding the actions that should be taken in different states of the LHD machine. To determine the optimal number, test training was conducted on the simulated LHD machine in the straight lane. The expected walking path is shown in Figure 6. In the five groups of experiments designed, the other parameters remained unchanged, and only the number of neurons was changed. The experimental parameters are shown in Table 3.

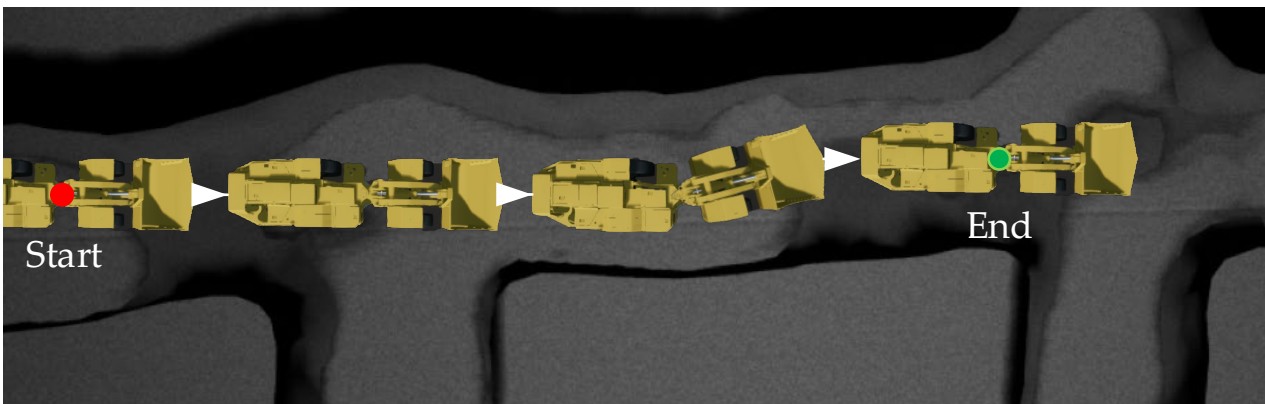

**Figure 6.** The optimal walking path of the LHD machine. The red point indicates the start location and the green point indicates the end location.

**Table 3.** Parameter settings of the different groups.

| Group | Batch Size | Learning Rate | Reward Discount | Memory Capacity | Neurons |
|-------|-----------|---------------|-----------------|-----------------|---------|
| 1 | 32 | 0.01 | 0.9 | 100 | 5 |
| 2 | 32 | 0.01 | 0.9 | 100 | 10 |
| 3 | 32 | 0.01 | 0.9 | 100 | 15 |
| 4 | 32 | 0.01 | 0.9 | 100 | 20 |
| 5 | 32 | 0.01 | 0.9 | 100 | 25 |

In the experiment, the action space was set as $\overline{A} = \{a_1, a_2, a_3, a_4\}$ straight, backward, left, and right, corresponding to $a_1$, $a_2$, $a_3$, and $a_4$, respectively. The state space was set as $\overline{S} = \{s_1, s_2, s_3, s_4\}$ speed, steering angle, distance from starting point, and distance from ending point, corresponding to $s_1$, $s_2$, $s_3$, and $s_4$, respectively. To avoid the LHD machine becoming too close to the tunnel, so that the LHD machine crosses the endpoint but is not considered to arrive endpoint, we set $\varepsilon = 3$ m, $\delta = 1.7$ m. and made sure there were positive rewards after the LHD machine takes the correct behavior, otherwise the training of the LHD would not converge; thus, the reward rules during training are shown in the formula (7):

$$\begin{cases} R = -1 \\ R = R - 1 & if\ D_l < \delta\ or\ D_r < \delta \\ R = R + 20 & if\ s_4 < \varepsilon \\ R = R - 10 & if\ crashed \end{cases} \tag{7}$$

where $D_l$ and $D_r$ are the distance from the left side and the distance from the right side, respectively.

### 3.1.2. Experimental Turning the LHD Machine Based on the DQN Algorithm

After determining the number of neurons, an intersection in the tunnel was chosen and steering training was conducted for the LHD machine. The expected path is shown in Figure 7. To speed up the training, the rewards and punishments related to the driving direction of the LHD machine were increased; the rewards were set as shown in Formula (8).

$$
\begin{cases}
R = (s_3' - s_3) \times 20 + (s_4 - s_4') \times 5 - 20 \\
R = R + 15 & if\ D_l > \delta\ and\ D_r > \delta \\
R = R + 200 & if\ s_4 < \varepsilon \\
R = R - 80 & if\ crashed
\end{cases}
\tag{8}
$$

where $s'$ represents the state of the last time.

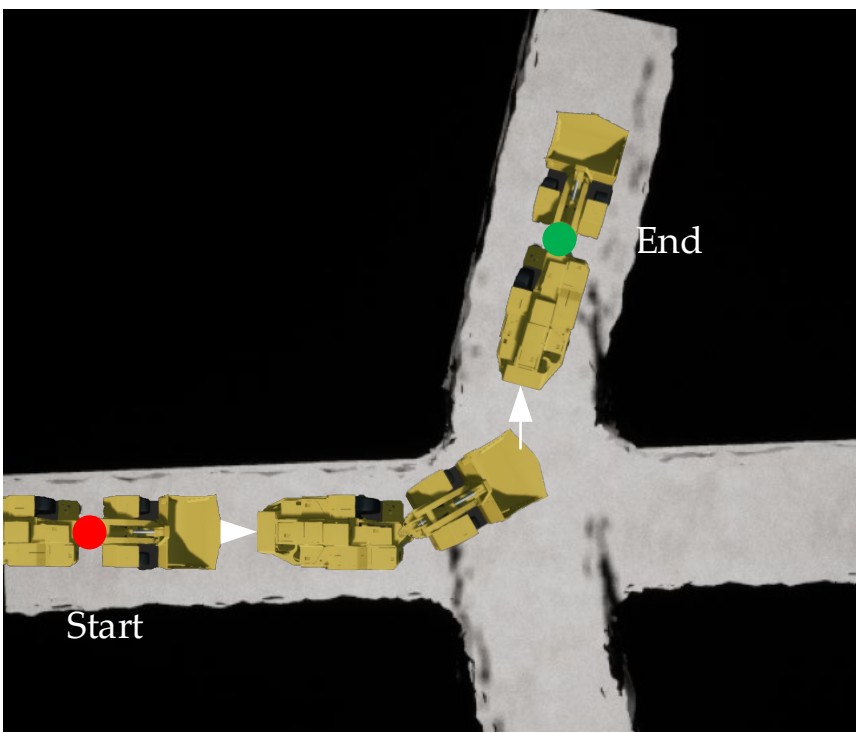

**Figure 7.** The expected path of DQN turning training. The red point indicates the start location, and the green point indicates the end location.

### 3.2. Experiment of LHD Machine Autonomous Walking Based on TCB-DQN

When using DQN to train the LHD machine to walk independently, the factor that most affected the training efficiency of the LHD machine was the ineffective exploration of the LHD machine in the straight tunnel. To reduce the ineffective exploration of the LHD machine, the LHD machine was controlled with the traditional control algorithm in the straight tunnel, because it is not necessary to turn left or right when the LHD machine runs in a straight tunnel. As shown in Figure 8a, the articulated angle $\gamma$ of the simulated LHD machine shall be maintained at 0 in the straight tunnel, there are no obstacles that can be detected within a certain range at the front, and $L_f$ is false. The tunnel can be detected on the left and right sides of the LHD machine, $L_l$ and $L_r$ are true. The distances from the tunnel $D_l$ and $D_r$ can be collected. When the distance is safe, the LHD machine only needs to go straight. When it is too close to one side of the tunnel, as shown in Figure 8b, the articulated angle is adjusted. When the LHD machine arrives at the intersection, as shown in Figure 8c, the radar status changes, and the left or right radar cannot detect objects. In this case, the control of the simulated LHD machine is handed over to the reinforcement learning, as shown in Figure 8d, and the DQN algorithm is used to determine the articulated angle of the LHD machine. With this method, DQN is only used to determine the $\gamma$. When

the radar status changes to undetectable objects on the front side and detected objects on the left and right side, the control of the simulated LHD machine is handed over to the control algorithm.

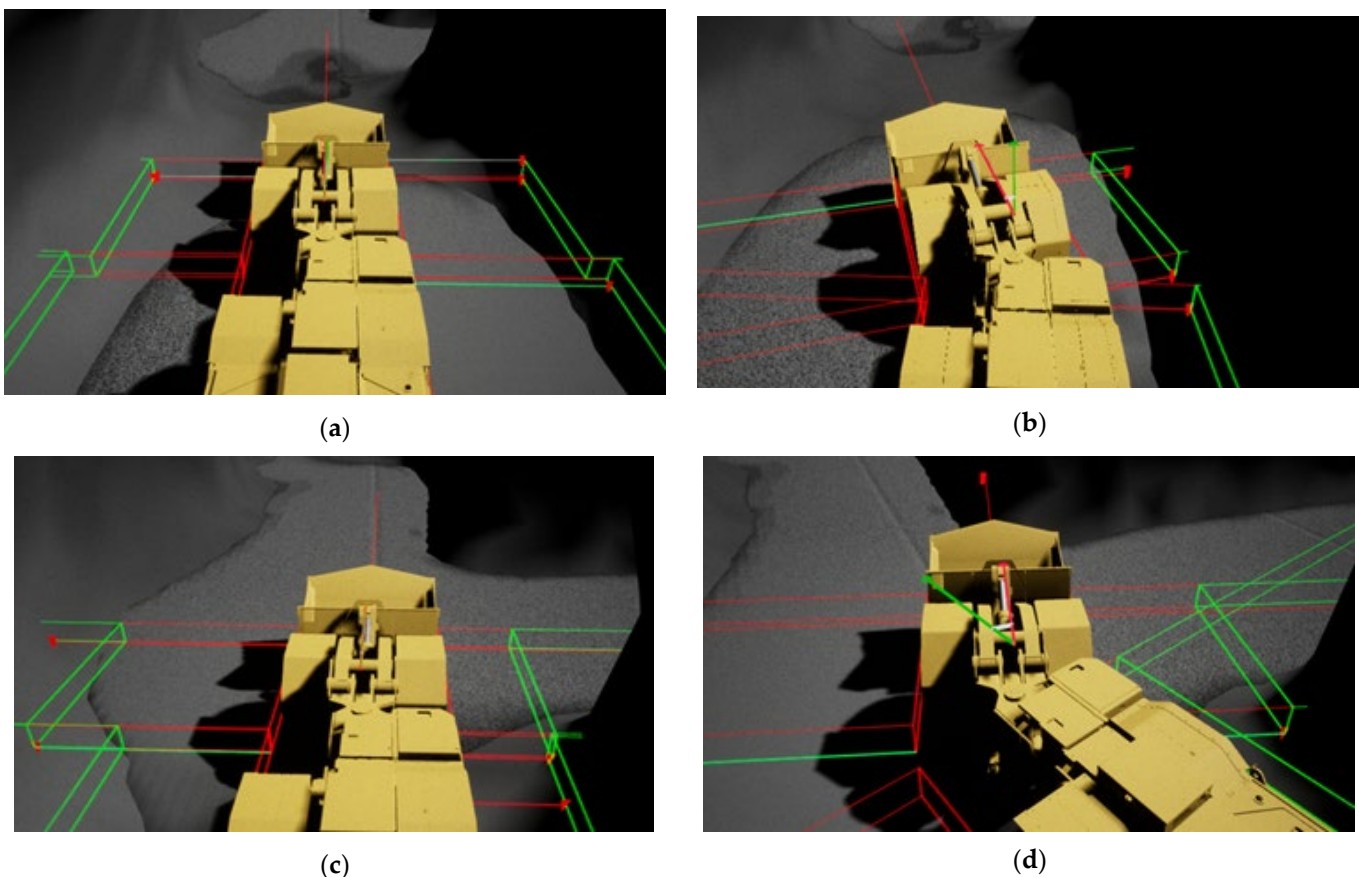

(**a**)

(**b**)

(**c**)

(**d**)

**Figure 8.** The monitoring condition of LHD machine walking in the tunnel. The green box indicates the detection range, the red line indicates the distance from the detection point to the scraper; (**a**) run in the straight tunnel, front radar status is false. Left and right radar status are true; (**b**) run in the straight tunnel, too close to the right; (**c**) run in the intersection, right radar status is false; (**d**) turn round, the radar status changes constantly.

According to the above ideas, a new training algorithm named Traditional Control Based DQN (TCB-DQN) is proposed, as shown in Algorithm 1.

---

**Algorithm 1.** TCB-DQN

---

1: **while** the LHD machine does not arrive endpoint, **do**
2:      **if** $L_f$ is false, $L_l$ is true, and $L_r$ is true, **then**
3:          Make the LHD machine move forward
4:          **if** $D_l$ and $D_r$ are safe **then**
5:              **if** $\gamma > 5$ **then**
6:                  Decrease $\gamma$ to 0
7:              **end if**
8:                  **if** $\gamma < -5$ **then**
9:                      Increase $\gamma$ to 0
10:                 **end if**
11:         **end if**
12:         **if** $D_l$ or $D_r$ are unsafe **then**
13:             **if** $D_l$ is unsafe **then**
14:                     Increase $\gamma$
15:             **end if**
16:                 **if** $D_r$ is unsafe **then**
17:                     Decrease $\gamma$
18:                 **end if**
19:     **end if**
20:     **else**
21:         Use DQN decide $\gamma$
22:         Make the LHD machine move forward
23:     **end else**
24:     **if** the LHD machine arrives endpoint **then**
25:             break
26:     **end if**
27: **end while**

---

3.2.1. Feasibility Verification Experiment of the TCB-DQN Algorithm

Due to the change in the algorithm, the parameters are also changed when using DQN. The neural network structure is shown in Figure 9, and the original action space $\overline{A} = \{a_1, a_2, a_3, a_4\}$ is changed to $\hat{A} = \{a_1, a_2\}$. $a_1$ and $a_2$ correspond to reducing the articulated angle and increasing the articulated angle, respectively. The original state space $\overline{S} = \{s_1, s_2, s_3, s_4\}$ becomes $\hat{S} = \{s_1, s_2, s_3, s_4, s_5\}$, $s_1$, $s_2$, $s_3$, $s_4$, and $s_5$, respectively, corresponding to the speed $v$, articulation angle $\gamma$, left distance $D_l$, right distance $D_r$, and distance from the endpoint $D_f$ of the LHD machine; in the experiment, $\varepsilon = 4$ m, when $D_f < \varepsilon$, it was considered to have reached the endpoint. The LHD machine can achieve $\gamma = 0$ if $-3 < \gamma < 3$, this can avoid driving route deviations caused by a small angle. The training rules for each step are shown in Equation (9). The expected path is shown in Figure 10.

$$\begin{cases} R = -1 \\ R = R + 20 & if\ s_5 < \varepsilon \\ R = R - 10 & if\ crashed \end{cases} \tag{9}$$

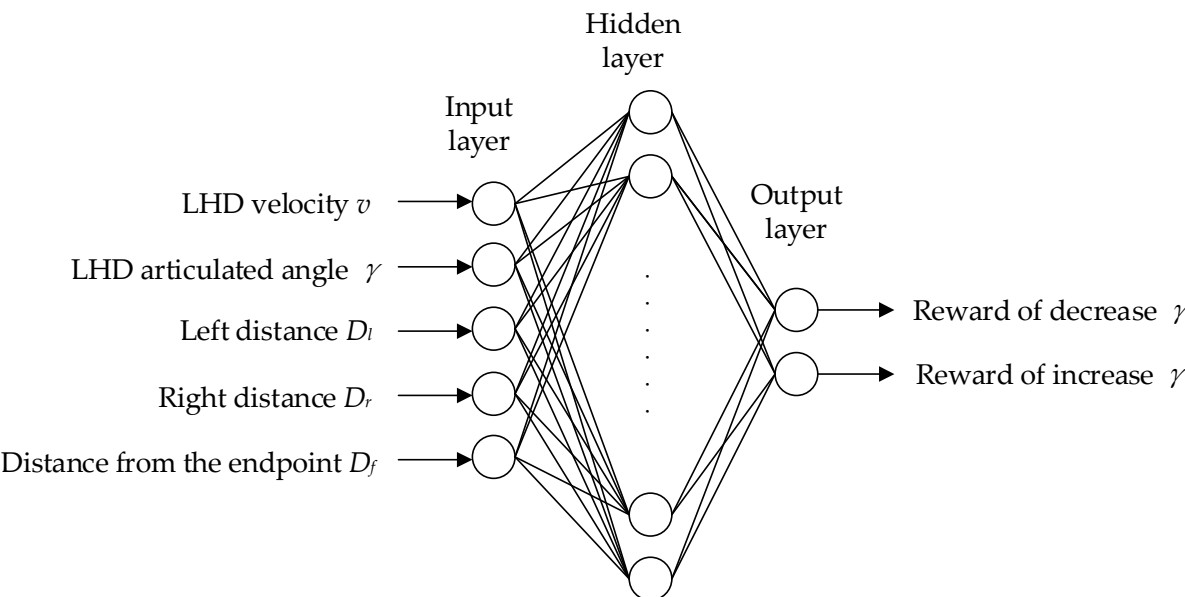

**Figure 9.** The structure of the neural network used in TCB-DQN. In the simulation environment, a detection box was launched from the side of the LHD machine, the first point that the box contacts was selected, and $D_l$ and $D_r$ are defined as the distance from this point to the LHD machine.

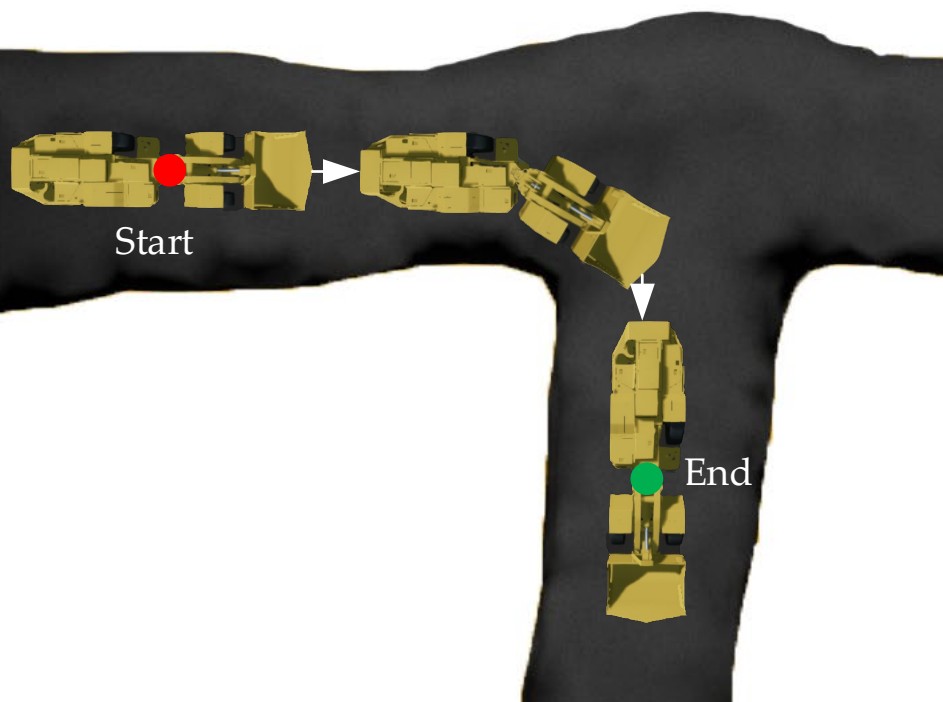

**Figure 10.** The expected turning path of the LHD machine based on TCB-DQN.

3.2.2. Experimental Design of TCB-DQN Robustness Verification

To further prove the good robustness of the TCB-DQN, the LHD machine was trained in a more complex tunnel, as shown in Figure 11. This condition meant that the LHD machine needed to travel a longer distance. If a reward is provided only when reaching the endpoint, the training process for the LHD machine would be extremely long. Therefore, a middle point was set on the way and add a distance $D_m$ from the middle reward point in the state space. At this time, the state spaces $\hat{S} = \{s_1, s_2, s_3, s_4, s_5, s_6\}$ $s_1$, $s_2$, $s_3$, $s_4$, $s_5$, and $s_6$ correspond to the speed $v$, articulation angle of the LHD machine respectively $\gamma$,

left side distance $D_l$, right side distance $D_r$, distance $D_m$ from middle reward point, and distance $D_f$ from final reward point. Set $\varepsilon_1 = 5$ m, $\varepsilon_2 = 4$ m, when $s_5 < \varepsilon_1$ and $s_2 = 0$, it is considered to reach the stage reward point; when $s_6 < \varepsilon_2$ and $s_2 = 0$, it is considered to reach the endpoint; the training rules are shown in Equation (10).

$$\begin{cases} R = -1 \\ R = R + 20 & if\ s_5 < \varepsilon_1\ and\ s_2 = 0 \\ R = R + 50 & if\ s_6 < \varepsilon_2\ and\ s_2 = 0 \\ R = R - 10 & if\ crashed \end{cases} \tag{10}$$

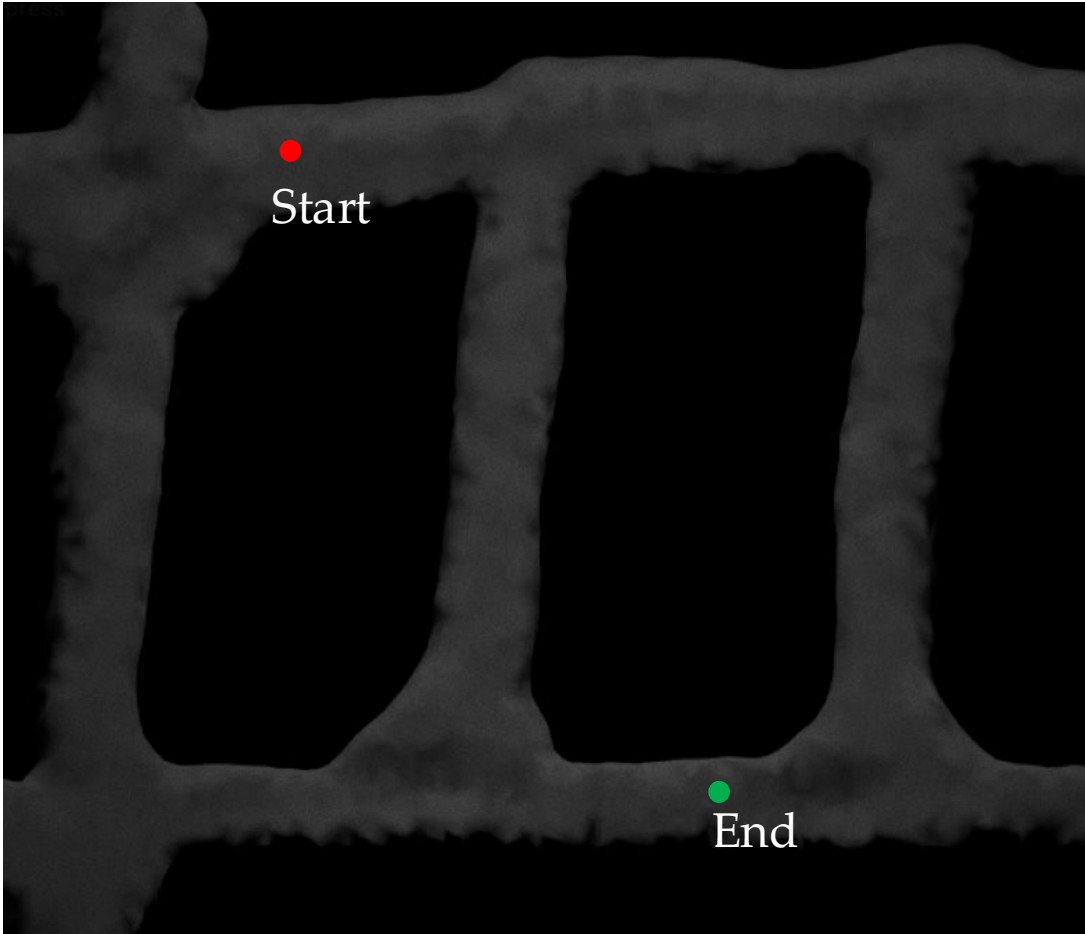

**Figure 11.** The complex underground scenario for training the LHD machine.

## 4. Experimental Results and Analysis

### *4.1. DQN Experimental Results and Analysis*

4.1.1. Experimental Results and Analysis of Determining the Number of DQN Neural Networks

The training results are shown in Figure 12 and Table 4, and the results of 100–140 experiments are taken as the experimental scores of different neurons in the final stage. Figure 13 is the boxplot of the results of 100–140 experiments in each group. Group3 had the highest average score and the lowest standard deviation. This proved that Group3 scored significantly higher than the other groups in the final stage.

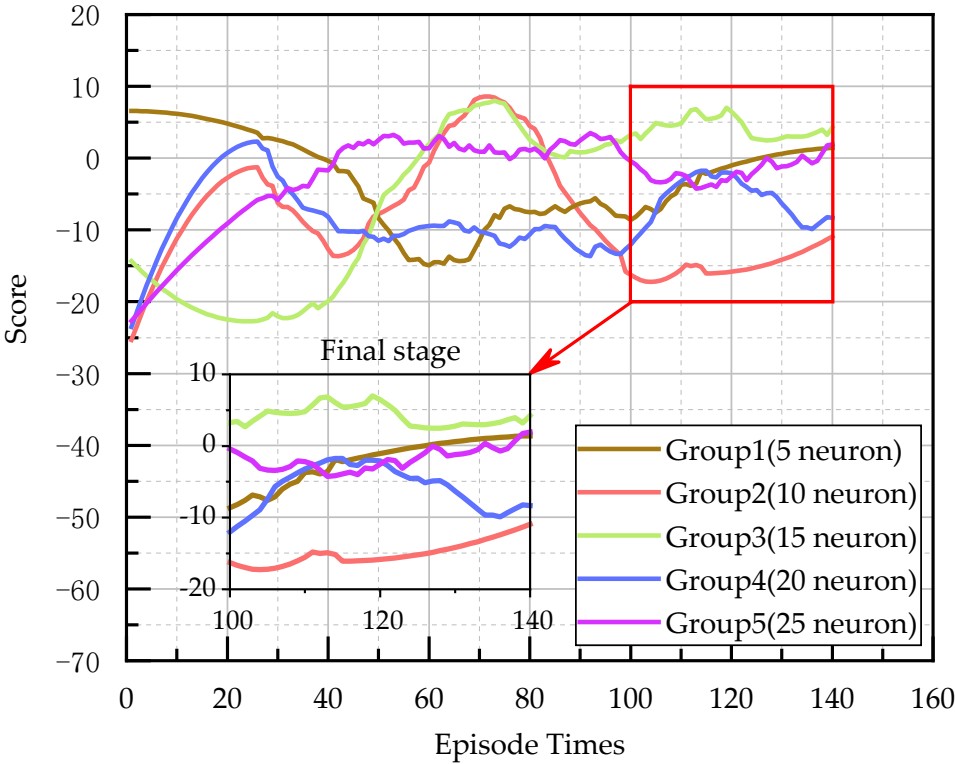

**Figure 12.** Comparison of the training results of different neurons numbers in the hidden layers.

**Table 4.** Descriptive statistics of 100–140 training scores.

| Group | Mean | Standard Deviation | Minimum | Median | Maximum |
|---|---|---|---|---|---|
| 1 | −1.58001 | 2.07777 | −5.11647 | −1.54344 | 1.82062 |
| 2 | −14.39721 | 1.09886 | −15.60397 | −14.70304 | −11.90487 |
| 3 | 3.08222 | 0.45751 | 2.67798 | 2.86441 | 4.21315 |
| 4 | −7.8594 | 1.7001 | −11.80333 | −7.15154 | −6.29695 |
| 5 | 0.39877 | 2.04473 | −1.75212 | −0.21574 | 4.82521 |

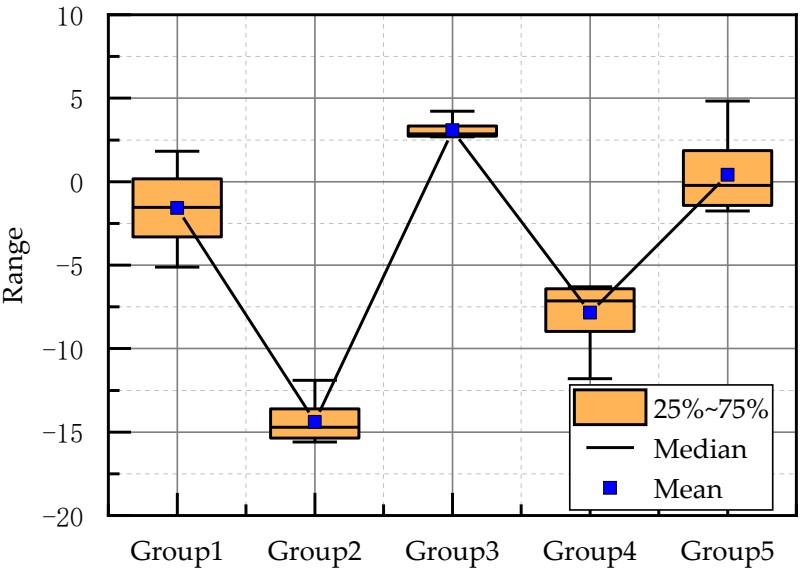

**Figure 13.** Comparison of 100–140 training scores.

To determine whether the autonomous driving action strategy of the LHD machine conformed to the expected trajectory, a further analysis of the actions taken by the LHD machine during the Group3 training was performed, and the actions taken by the LHD machine in each event in the last stage of the training were recorded. As shown in Table 5, the LHD machine initially carried out a straight operation. When it runs to approximately step 5, the LHD machine carried out calibration and was far away from the right tunnel. When it was roughly running in the center of the tunnel, it ran to the right again to ensure that the direction of the LHD machine was consistent with the center of the tunnel, which was roughly consistent with the expected walking path of the LHD machine; and the score of each event also reflected a good overall operation. Therefore, the experiment showed that when the number of neurons was set to 15, the experiment could achieve good results.

**Table 5.** The action and score of the LHD machine with a 15-neuron DQN algorithm.

| Episode | Operation Steps | | | | | | | | Score |
|---|---|---|---|---|---|---|---|---|---|
| | Step 1 | Step 2 | Step 3 | Step 4 | Step 5 | Step 6 | Step 7 | Step 8 | |
| Manual | ↑ | ↑ | ↑ | ↑ | ↑ | ← | → | → | / |
| 1 | ↑ | ↑ | ↑ | ↑ | ← | → | → | ← | 11.00 |
| 2 | ↑ | ↑ | ↑ | → | ↑ | ← | → | ← | 11.00 |
| 3 | ↑ | ↑ | ↑ | ↑ | ↑ | ↑ | ← | → | 12.00 |
| 4 | ↑ | ↑ | ↑ | ↑ | ↑ | → | ↑ | ← | 11.00 |
| 5 | ↑ | ↑ | ↑ | ↑ | ↑ | ← | → | ← | 12.00 |
| 6 | ↑ | ↑ | ↑ | ← | ↑ | → | ← | → | 11.00 |
| 7 | ↑ | ↑ | ↑ | ↑ | ← | ↑ | ← | → | 12.00 |
| 8 | ↑ | ↑ | ↑ | ↑ | ↑ | ← | → | ← | 12.00 |
| 9 | ↑ | ↑ | ↑ | → | ↑ | ← | ← | ↑ | 11.00 |
| 10 | ↑ | ↑ | ↑ | ↑ | ↑ | ← | → | ← | 12.00 |

### 4.1.2. Experimental Results and Analysis of the LHD Machine Turning Based on DQN

The training results are shown in Figure 14. When the training reached 450 iterations, the score of an event tended to be stable and was larger than the score at the beginning of the training. In the observation of the training process of the LHD machine, it was found that although the LHD machine could reach the endpoint eventually, it still took unnecessary actions during the DQN training. Events with higher scores could still be seen, which were directly related to the unnecessary actions taken by the LHD machine. As shown in Table 6, in the fifth step of each event, there was a straightforward operation. If the operation is manual, there will be no redundant operation in the fifth step. Although the actions and scores taken in the last stage of training were relatively stable, there is still room for improvement.

**Table 6.** The actions and scores of the LHD machine in the final stage.

| Episode | Operation Steps | | | | | | | Score |
|---|---|---|---|---|---|---|---|---|
| | Step 1 | Step 2 | Step 3 | Step 4 | Step 5 | Step 6 | Step 7 | |
| Manual | ↑ | ↑ | ↑ | ↑ | ← | ← | ← | / |
| 1 | ↑ | ↑ | ↑ | ← | ↑ | ← | ← | 81.47 |
| 2 | ↑ | ↑ | ↑ | ← | ↑ | ← | ← | 81.2 |
| 3 | ↑ | ↑ | ↑ | ← | ↑ | ← | ← | 80.43 |
| 4 | ↑ | ↑ | ↑ | ← | ↑ | ← | ← | 80.78 |
| 5 | ↑ | ↑ | ↑ | ← | ↑ | ← | ← | 80.5 |

When using the DQN algorithm for training, even in the straight tunnel, the LHD machine will carry out redundant exploration actions, resulting in many meaningless commands. Although increasing the number of trainings may improve this situation, an event may be up to dozens of timesteps in complex cases, and each timestep has four action

choices. Even if global optimization can be achieved through high-intensity training, this would also consume a lot of energy and material resources. Therefore, the training efficiency is also one of the important indicators in evaluating the advantages and disadvantages of the algorithm.

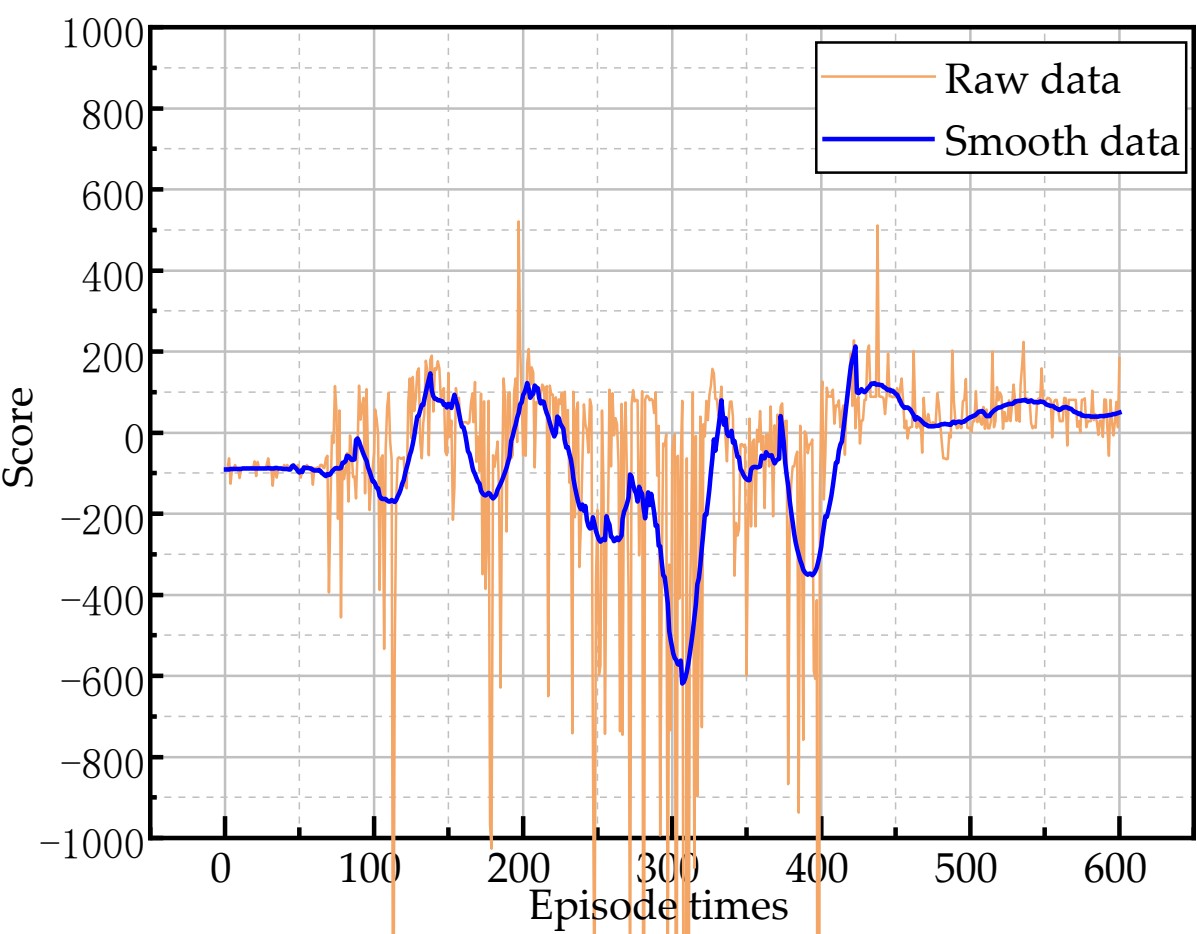

**Figure 14.** The DQN training results.

*4.2. Experimental Results and Analysis of TCB-DQN*

4.2.1. Experimental Results and Analysis of the Preliminary Application of the TCB-DQN Algorithm

The training results are shown in Figure 15. Since the reinforcement learning algorithm was only used to control turns, the convergence speed was significantly faster than that of DQN. There was a convergence trend when DQN was trained only about 450 times, while TCB-DQN was relatively stable when TCB-DQN was trained about 100 times, and the training time was reduced by 77%.

In addition, the action list taken by the LHD machine was observed, and the action in the final stage was noted, as shown in Table 7, which was almost the same as the expected driving mode. Even with a manual operation, it will choose to set the articulation angle to the maximum when turning right. However, the TCB-DQN algorithm will be more time-consuming than manual operation, as manual operation can increase the articulation angle and speed up at the same time, while TCB-DQN has to adjust the articulated angle and travel operation step by step.

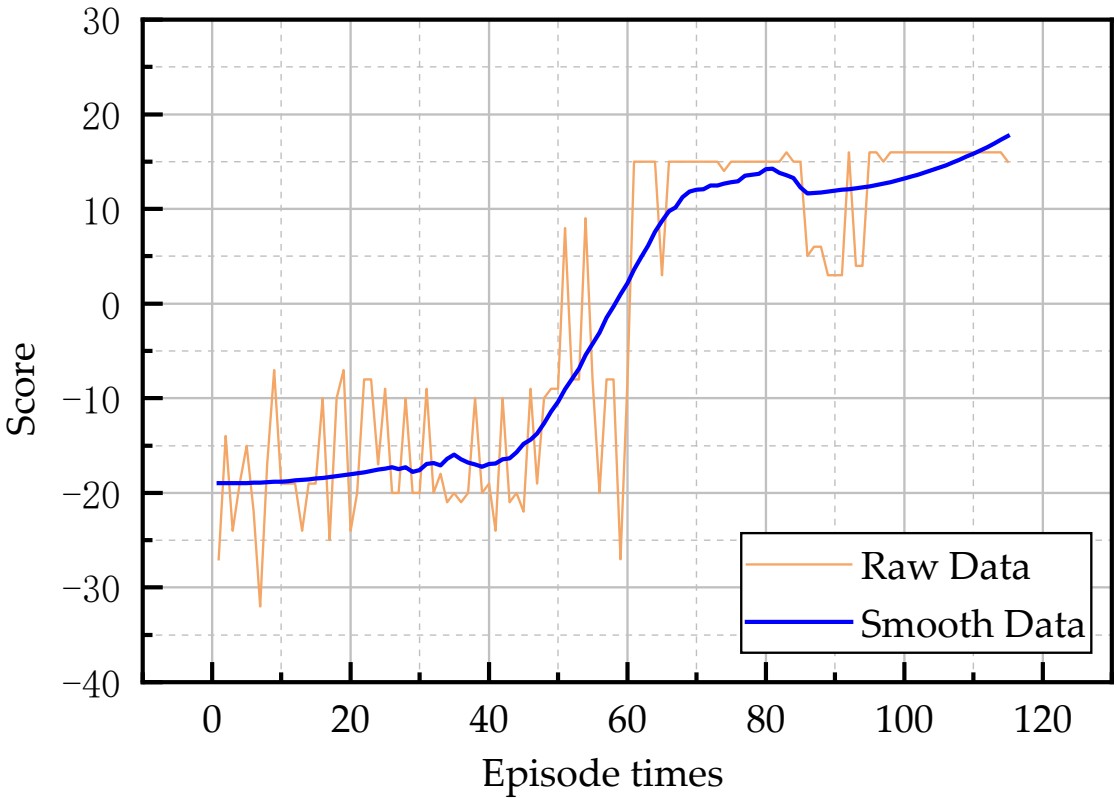

**Figure 15.** The TCB-DQN training results.

**Table 7.** The actions and scores of the LHD machine in the final stage of turning based on TCB-DQN.

| Episode | Operation Steps | | | | Score |
|---|---|---|---|---|---|
| | Step 1 | Step 2 | Step 3 | Step 4 | |
| Manual | → | → | → | / | / |
| 1 | → | → | → | / | 16 |
| 2 | → | → | → | / | 16 |
| 3 | → | → | → | / | 16 |
| 4 | → | → | → | / | 16 |
| 5 | → | → | → | / | 16 |
| 6 | → | → | ← | → | 15 |
| 7 | → | → | → | / | 16 |
| 8 | → | → | → | / | 16 |
| 9 | → | → | → | / | 16 |
| 10 | → | → | → | / | 16 |

4.2.2. Experimental Results and Analysis of TCB-DQN Robustness Verification

The training results are shown in Figure 16. It can be seen from the figure that when the training reached about 250 times, the score fluctuated around 20 points, indicating that the LHD machine had learned how to reach the middle stage point through the first intersection. When it reached about 1800 times, the LHD machine had successfully learned how to pass the second intersection. When it reached about 2000 times, the score of the LHD machine tended to converge.

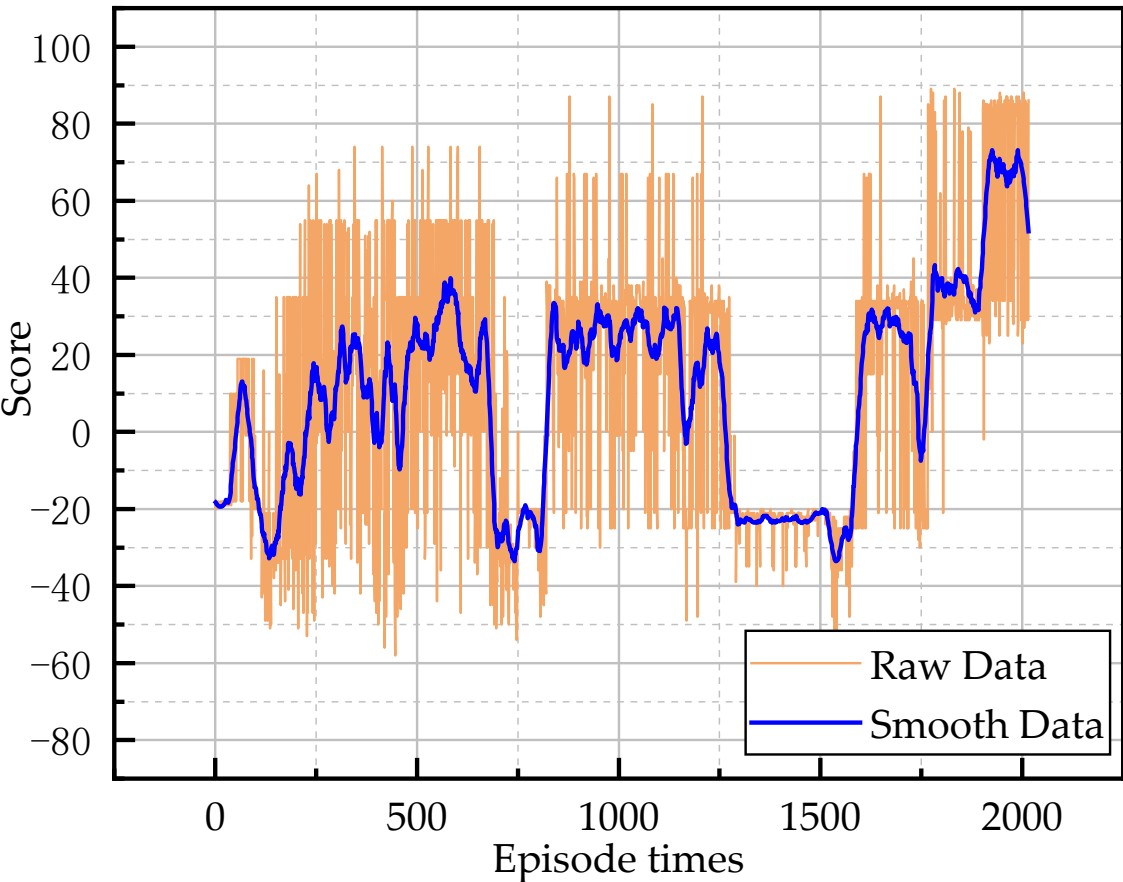

**Figure 16.** TCB-DQN training results under complex tunnel conditions.

Figure 17 shows the changes in the LHD machine's running track during the whole training process. At first, the LHD machine explored in the direction of the first intersection. After exploration, it was found that only turning right can reach the middle stage point. Then, after passing through the straight tunnel using the control strategy, the LHD machine explored the direction of the second intersection. Due to the large curvature of the second intersection, the LHD machine explored the turning mode many times. Finally, it could determine a route to keep the score stable. At the same time, the DQN algorithm combined with the conditions of the Equation (10). was used to train the LHD machine, but 1200 training iterations could not make the LHD machine successfully reach the second intersection, which shows that TCB-DQN could effectively reduce the ineffective exploration action of the DQN algorithm, and the training efficiency was significantly better than the DQN algorithm. It could still realize relatively stable autonomous walking of the LHD machine under complex conditions. The algorithm has strong robustness.

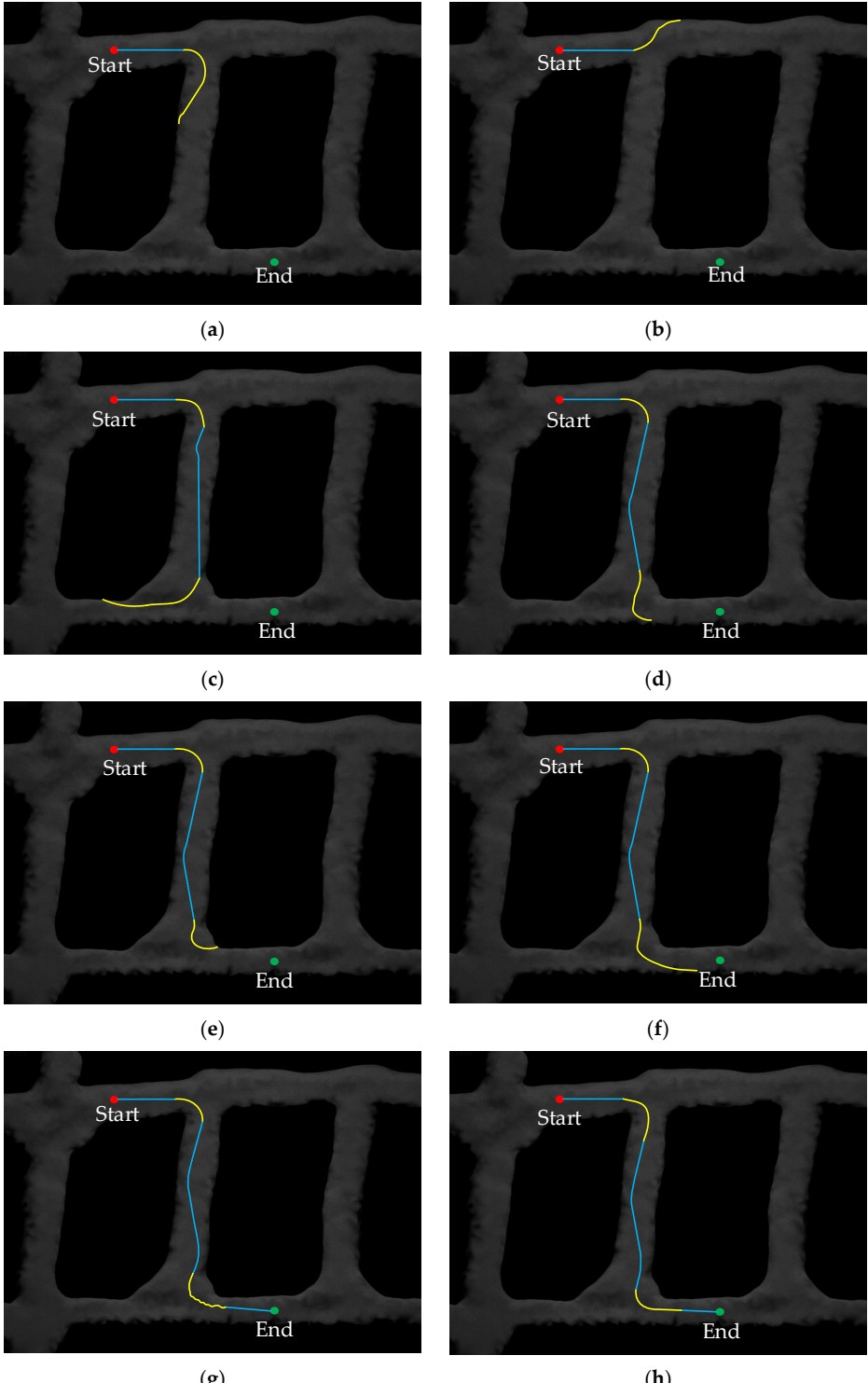

**Figure 17.** Different stages of TCB-DQN training, the blue line indicates that the LHD machine is controlled by the traditional control algorithm, and the yellow line indicates that the LHD is controlled by the DQN algorithm. (**a**) LHD machine makes a right turn exploration at the first turning point; (**b**) LHD machine makes a left turn exploration at the first turning point; (**c**) LHD machine makes a right turn exploration at the second turning point; (**d**) LHD machine makes a left turn exploration at the second turning point; (**e**) LHD machine turned late and hit the tunnel; (**f**) LHD machine explore a method for turning; (**g**) although the LHD machine reached the endpoint successfully, the turn at the second turning point was not smooth enough; (**h**) LHD machine final training track.

## 5. Conclusions and Discussion

This paper aimed to realize the autonomous walking of an underground LHD machine in a tunnel, and built an underground working scenario based on a real tunnel and LHD machine in UE4. In this scenario, the feasibility of reinforcement learning in the autonomous walking control of the LHD machine was verified, and the experimental effect of the DQN algorithm in multi-state reinforcement learning training was further verified. In the experiment, inspired by reflective navigation and the DQN algorithm, a new training framework TCB-DQN was proposed, which combines the traditional control algorithm and DQN algorithm. This algorithm could realize the autonomous walking of the LHD machine faster and better. The experiments showed that TCB-DQN could avoid unnecessary exploration of the LHD machine in reinforcement learning training, and shortened the training time by 77% compared with the pure DQN algorithm. Finally, the robustness of the TCB-DQN algorithm was verified in a complex tunnel. The experiments showed that even if the training environment was changed to a more complex tunnel, a feasible LHD machine control method could still be obtained by using the TCB-DQN algorithm. The innovations of this paper are as follows:

(1)  It was proven that the reinforcement learning algorithm is feasible for realizing the autonomous walking of a LHD machine in a definite scene.

In previous studies, research on the walking of a LHD machine often focused on the control algorithm. The advantage of a control algorithm is that it is relatively stable. Around a prepared standard, the LHD machine can move to the greatest extent according to this standard. However, the prepared standard is also formulated with the participation of people, which inevitably leads to some biases. According to the preferences of different makers, the standards may be different. The use of the reinforcement learning method can endow the LHD machine itself with the capability for exploration of the environment, without giving standards in advance. After repeated training in large quantities, it can often achieve better results.

(2)  A TCB-DQN algorithm that integrates the traditional control algorithm and DQN was proposed.

Combining the advantages of the traditional control algorithm and DQN, a TCB-DQN algorithm was designed. Using this algorithm can greatly reduce the unnecessary exploration behavior of a simulated LHD machine; thus, it can greatly reduce the time of using the algorithm to train the simulated LHD machine, and use DQN to explore the best driving mode at key positions. Experiments showed that TCB-DQN can achieve an effect similar to, or even surpassing, manual operation.

(3)  Autonomous walking of the simulated LHD machine in the simulation environment was realized, and this laid the foundation for parallel driving of the LHD machine.

To make a LHD machine unmanned, starting with the walking of the LHD machine, how to realize the autonomous walking of the LHD machine simply and efficiently must be considered. The most direct method is to install lidar and various mechanical sensors on a real LHD machine. However, the transformation of a LHD machine is a difficult problem, and the LHD machine and various equipment are easily damaged in the training process. Therefore, modeling of the real scenario and the LHD machine is carried out, Training the LHD machine in the virtual environment can complete the training effect that cannot be achieved in the real scenario in a short time. After the training is successful, the results will be transferred to the real LHD machine controller, which can realize the autonomous walking of the LHD machine, with the advantages of low cost and high efficiency.

Tunnels are often sprayed with concrete, which makes the tunnel wall appear full of folds. In the process of modeling, to make the tunnel look more real, the tunnel wall model is artificially adjusted. The folds and bulges on the tunnel wall should have a collision volume, but it is often found that some folds and bulges lose collision volume in the simulation environment, resulting in the occurrence of formwork penetration; this

is because the collision box does not perfectly cover the tunnel model. In future research, the number convex hulls can be adjusted to make the tunnel model collision more real; but at the same time, the problem of computer resource allocation should be considered to balance quality and performance.

Reinforcement learning has been mainly divided into two parts: one is value-based, and the other is policy-based. The former provides a deterministic action at each step, and the latter provides the probability of different actions being selected at each step. They have their advantages and are developing in the direction of gradual integration. In the experiment, solely a value-based DQN algorithm was used in reinforcement learning. In future work, other algorithms of reinforcement learning can be used to train the LHD machine, which may achieve better results.

**Author Contributions:** S.Z.: data curation, investigation, software, writing-original draft, writing—review and editing; L.W.: conceptualization, formal analysis, funding acquisition, writing—review and editing; Z.Z.: methodology, resources, writing—review and editing; L.B.: formal analysis, funding acquisition, supervision, writing—review and editing. All authors have read and agreed to the published version of the manuscript.

**Funding:** This study was funded by the National Key R&D Program of China, 2019YFC0605300.

**Institutional Review Board Statement:** Not applicable.

**Informed Consent Statement:** Not applicable.

**Data Availability Statement:** Not applicable.

**Acknowledgments:** The authors gratefully acknowledge the funders and all advisors and colleagues who supported our work.

**Conflicts of Interest:** The authors declare no conflict of interest. The funders had no role in the design of the study; in the collection, analyses, or interpretation of data; in the writing of the manuscript, or in the decision to publish the results.

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
