# Peer review of "Study on the Autonomous Walking of an Underground Definite Route LHD Machine Based on Reinforcement Learning"

_applsci, doi:10.3390/app12105052_

Round 1
Reviewer 1 Report
This is an interesting paper that presents an study on autonomous walking of an underground vehicle using reinforcement learning. The authors have built a simulation environment including LHD machine modeling, which was used to train a proposed framework called TCB-DQN. The manuscript have a detailed description of model construction and algorithm design, as well as experimental results and discussion. Here are some comments:
1. Define the LHD acronym when used in the Abstract and main body of the manuscript.
2. Make all your figures consistent: size, appearance (use of grid) and labels of axis.
3. Check for multiple spelling issues in tables and figures.
4. The most important comment would be that the manuscript have a more engineering report feeling than a scientific journal contribution. Please, in the introduction state with more clarity the main scientific contributions done in this work, and how it improves the state-of-the-art in this area.
Reviewer 2 Report
Please if possible avoid to use "We" in a scientific paper, porefer the use of passive voice (e.g. line 85, 93, 109, 194, 196, 296, 314, 325, 328, 329, 350, 488, 515, 565, 566)
Line 99 replace the ";" with a .""(...LHD machine; In the case of steering, TCB-DQN can......)
Line 103: please do not use capital letter after a comma or replace with a "." (....simulation environment, In the third section....)
Lines 119 and 123: please do not use capital letter after a comma or replace with a "." (...is shown in Figure 1(c), It can 119 be seen that the ground of..) (..flattened in UE4, The final tunnel is shown in Figure 1(d).)
Line 124 check the use of capital letters and replace as follow: (a) Original point cloud data...
Figure 1(a) can be added a legend for the colors?
Table 1 Use capital letter for the title of columns "Acceleration" "Deceleration"
Move the Figure 4 before Table 2 following the order used in the text to intruce the two elements.
Please check the formula 7 at line 245 line 247 some symbols is probably not well visualised .
Line 268: please do not use capital letter after a comma or replace with a "." (....interval, This is bound to produce errors.)
Move the Figure 6 before Table 3 following the order used in the text to intruce the two elements.
Lines 329-331, please review the sentences they are not clear( " We expect that when the LHD machine runs in a straight tunnel, it is not necessary to turn left or right. When the simulated LHD machine travels in the straight tunnel, as shown in Figure 8(a).)
Line 394: please do not use capital letter after a comma or replace with a "." (....of Q-learning is immediate, That...)
Line 421 replace "group3" with "Group3"
Line 425: please do not use capital letter after a comma or replace with a "." (...of the tunnel, Run to the right...)
Move the Figure 15 before Table 6 following the order used in the text to intruce the two elements.
Line 451 use capital letter for table 6 caption (Table 6. The actions and scores were taken by the LHD machine in the final stage.)
Line 509: please do not use capital letter after a ":" or replace with a "." (....is further verified; In the....)
Line 556: please do not use capital letter after a ":" or replace with a "." (....formwork penetration, This is....)
Reviewer 3 Report
In general, I think the paper is good because it is very close to the practical use through the consideration of real cases and related software to perform the simulation. The solution method is quite good, but more literature review (especially for the method development) might need necessary, e.g., there could be many routing-related studies that uses DQN, but there were no much discussions about them in the manuscript. Please find my detailed review below:
[Comment 1] Novelty
[Subcomment 1a] When mentioning the contribution of the study, the authors need to clearly compare their research's novelty with other previous specific studies (please mention those study and how they are different). If necessary, please compare them in a table while listing the compared characteristics.
[Subcomment 1b] Please compare the DQN design with other routing-related studies. The authors need to emphasize the novelty of their DQN design and why they design their DQN as it is (the authors should explain the design process until they finalized their design as presented in the manuscript).
[Comment 2] Proposed method
[Subcomment 2a] Instead of providing a general design on the reinforcement learning (RL) in Section 3.1 and the initial DQN design in Section 3.2, I suggest the authors to just present the final algorithm to avoid confusion. The details are actually explained in Section 3.3.1, but reading only Section 3.1 cause a lot of confusion, as follows (and I do not think it is worth to explain it in Section 3.1 to avoid redundancy. (If the authors remove all explanations about general RL and the initial DQN design with 4 movement directions, the following subsubcomments in this subcomment can be ignored):
- (Section 3.1) Please state how the authors conduct the parameter tuning, e.g., distances, reward, etc.
- Please mention all exact values of the parameters, including the large reward given when the LHD machine reaches the endpoint.
- In line 241, they mention that they set -20 for each step taken and give a lot of rewards when reaching the endpoint. Then, where do the authors use Equation (7)? It should not refer to reaching the endpoint, because there is a certain "change" from l_1. Please give a short introduction about Equation (7) to avoid confusion. I would suggest the authors replace it with Equation (9) directly.
- (lines 247-248) It seems that the authors refer to the distance when mentioning about the "change". Please be specific and clear here.
- Is the LHD splice angle controllable? If yes, should not it be included as the actions (in the output layer of Figure 5)? If the authors place it at the input layer, I believe that its detailed relationship (calculation) with the movement directions should be addressed.
[Subcomment 2b] (another similar review comment with the subcomment above) Is there any reason why the authors present the first concept (Figure 5), before the second (Figure 9)? I was confused when trying to understand Figure 5 (about how to understand that the movements are different from the angle setting). I suggest the authors directly present the second concept only. If the authors insist on presenting the first concept as well, they might need to provide comparison between both methods in the numerical experiments (which I do not think would be possible, because the first concept is not implementable).
[Subcomment 2c] (line 273) Please add a reference showing that such reinforcement learning design was known as one of the best design (with best performance or as the state-of-the-art method).
[Subcomment 2d] Is it possible to detect any L_l and L_r from any distance? If not, then the machine might move zigzag when passing a straight tunnel. Please state the real hardware characteristic used for the detection.
[Subcomment 2e] Please be specific on the extent of large and small angle in Algorithm TCB-DQN.
[Comment 3] Numerical experiment
Please differently mark parts in Figure 18 that were produced by the traditional control method and by the control algorithm.
[Comment 4] Reference
[Subcomment 4a] (line 226) If the authors want to use the Cliffwalk example, the authors need to add the reference because not everyone is aware with this example.
[Subcomment 4b] Instead of using Atari and Cliffwalk, the authors need to use route finding-related studies using reinforcement learning (which are more related to the authors' study). Please provide the references.
[Comment 5] Writing quality
Please revise mistyped words, e.g., MemoriyCapacity (Table 3).
